# Efficiency of the Enzymatic Conversion of Flavone Glycosides Isolated from Carrot Leaves and Anti-Inflammatory Effects of Enzyme-Treated Carrot Leaves

**DOI:** 10.3390/molecules28114291

**Published:** 2023-05-24

**Authors:** Joo Tae Hwang, Hye Jin Kim, Jin Ah Ryuk, Dong Ho Jung, Byoung Seob Ko

**Affiliations:** Herbal Medicine Research Division, Korea Institute of Oriental Medicine, Daejeon 34054, Republic of Korea; scorpion4119@gmail.com (J.T.H.); kimhyejin43@kiom.re.kr (H.J.K.); yukjinah@kiom.re.kr (J.A.R.); jdh9636@kiom.re.kr (D.H.J.)

**Keywords:** carrot leaves, *Daucus carota*, enzymatic conversion, quantitative optimization

## Abstract

In traditional oriental medicine, carrots (*Daucus carota* L.) are considered effective medicinal herbs; however, the use of *D. carota* leaves (DCL) as therapeutic agents has not been explored in depth. Therefore, we aimed to demonstrate the value of DCL, generally treated as waste while developing plants for wide industrial availability. Six flavone glycosides were isolated and identified from DCL, and their constituents were identified and quantitated using an NMR and HPLC/UV method, which was optimized and validated. The structure of chrysoeriol-7-rutinoside from DCL was elucidated for the first time. The method exhibited adequate relative standard deviation (<1.89%) and recovery (94.89–105.97%). The deglycosylation of DCL flavone glycosides by Viscozyme L and Pectinex was assessed. Upon converting the reaction contents to percentages, the luteolin, apigenin, and chrysoeriol groups showed values of 85.8, 33.1, and 88.7%, respectively. The enzyme-treated DCL had a higher inhibitory effect on TNF-α and IL-2 expression than that of the carrot roots or carrot leaves without enzyme treatments. These results highlight the importance of carrot leaves and could be used as baseline standardization data for commercial development.

## 1. Introduction

Carrot (*Daucus carota*), an important vegetable consumed by humans worldwide, is a vital source of nutrients and contains large quantities of carotenoids [1,2], such as provitamin A, *β*-carotene, and lutein. Carrots have long been used as antifungal, antibacterial, and nephroprotective agents [3,4] owing to their rich nutrient contents, including essential oils. In ancient Rome, carrots were used for treating pain and for detoxification. Furthermore, in traditional oriental medicine, carrots are considered effective remedies for constipation, dysentery, anemia, bladder inflammation, and measles; they also prevent kidney stones [5].

Carrots originated in Afghanistan and were first cultivated in China in the 13th and 14th centuries and in Korea in the 16th and 17th centuries [6]. However, only the roots have been used as therapeutic agents, and the use of carrot leaves has not been explored. In fact, carrots are mainly used as an essential food source rather than a therapeutic agent. The leaves are discarded and sometimes used to feed livestock in the private sector; however, with the mass production of quality feed, carrot leaves are not used for feed. 

Recent reports have revealed some chemical components and biological activities that can gradually increase the value of carrot leaves over that of carrot roots. According to previous reports, *β*-carotene content in carrot leaves is equivalent to or higher than that in carrot roots, even after heat treatment. Furthermore, essential oil content is significantly higher in the leaves than in the roots [6]. The total phenolic content is twice as much in the leaves as in the roots, and the antioxidant activity of water extracts of leaves is also twice as strong as that of the roots [7]. 

Furthermore, there are reports of an abundant content of flavones, such as luteolin and apigenin, in the leaves, but these have not been compared with the components in carrot roots [8]. In fact, luteolin and apigenin are mostly present in green leafy spices. The most extensively studied effects of flavones are their anti-inflammatory and anticancer activities [9,10]. Moreover, the intracellular penetration and activity of the aglycones of flavone glycosides, in which the sugar is removed, are known to be increased [11,12]. We believe that determining the exact flavone compounds in carrot leaves and roots and verifying their anti-inflammatory properties will aid in identifying novel applications of carrot leaves. The primary focus of this study was to demonstrate the value of the carrot leaves, which are generally treated as waste while developing plants for wide industrial availability.

In this study, six flavone glycosides (**1–6**) were isolated and identified from the n-BuOH layer of carrot leaves during standardization of methods for the separation of bioactives. In addition, the flavone glycosides were treated with Viscozyme and Pectinex, and three hydrolyzed flavones were isolated and quantitatively analyzed (Figure 1). The anti-inflammatory activities of carrot roots (DCR) and leaves (DCL) and enzyme-treated DCL were also investigated.

## 2. Results and Discussion

### 2.1. Structure Elucidation of the Isolated Compounds

Six flavone glucosides (**1**–**6**) were isolated from DCR during the standardization of methods for the separation of bioactives by repeated separation via semi-preparative high-performance liquid chromatography (HPLC) using an ODS gel column. The structures of all the isolates were identified and determined based on their spectroscopic data, such as their ^1^H and ^13^C nuclear magnetic resonance (NMR), including the 2D NMR (^1^H-^1^H correlated spectroscopy [COSY], heteronuclear multiple quantum coherence [HMQC], heteronuclear multiple bond correlation [HMBC]) and electrospray ionization mass spectrometry (ESI/MS) spectra, and compared with previously reported data.

Compound **1** was collected as a brown gel, and its molecular weight (594) was determined based on the ESI/MS ion peaks at *m/z* 595.0 [M + H]^+^ and 592.9 [M − H]^+^. In the ^13^C NMR spectrum, 27 resonances were observed, which were attributed to one CH_3_, one CH_2_, sixteen CH, and eight quaternary carbon atoms. Among the carbon resonances, the signal at *δ*_C_ 182.6 (C-4) indicated a ketone, and the chromophore site, C-3, *δ*_C_ 102.9, demonstrated that it was a flavone. It was also revealed at *δ*_H_ 6.56 (1H, s, H-3) in the ^1^H NMR spectrum. In addtion, the aromatic protons in the ABX system were observed at *δ*_H_ 7.37 (2H, s, H-2′, and H-6′) and 6.90 (1H, s, H-5′) of the C-ring and at *δ*_H_ 6.70 (1H, s, H-9) and 6.48 (1H, s, H-7) of the A-ring. Therefore, the aglycone of **1** was determined to be luteolin. Furthermore, two anomeric proton signals at *δ*_H_ 5.01 (1H, s, H-1″) and 4.69 (1H, s, H-1‴) demonstrated the *α*-rhamnopyranosyl and *β*-glucopyranose moiety, respectively. Furthermore, these anomeric protons were verified by proton–carbon correlations in the HMBC spectrum as H-1″/C-8 and H-1‴/C-6″, respectively. In the ^1^H-^1^H COSY spectrum, the proton–proton and proton–carbon as H-5″/H-6″ and H-4″/C-6″ correlations in the HMBC, respectively, indicated the arrangement of the *α*-rhamnopyranosyl and *β*-glucopyranose moieties as rutinosides. Therefore, compound **1** was determined as luteolin-7-rutinoside (scolymoside) based on the abovementioned experimental results and previously reported literature [13]. 

For compound **2**, which was collected as a brown powder, the ESI/MS ion peaks were observed at *m/z* 449.38 [M + H]^+^ and 447.80 [M − H]^+^, implying the loss of a rhamnose moiety compared with that in compound **1**, which has a luteolin aglycone moiety. In the ^13^C NMR spectrum, 21 resonances were observed. As in the NMR spectrum of **1**, the resonances of a glucose moiety at *δ*_C_ 77.7 (C-5″), 76.9 (C-3″), 73.7 (C-2″), and 70.1 (C-4″) and CH_2_ carbon at 61.2 (C-6″) had one anomeric proton at *δ*_C_ 100.5 (C-1″). This anomeric carbon was also assigned by correlation with a proton in the HMBC spectrum as H-1″ (*δ*_H_ 5.04)/C-8 (*δ*_C_ 100.5). Thus, compound **2** was determined as luteolin-7-O-glucoside (cynaroside) after comparing it with that in previous reports [14,15].

Compound **3** was collected as a brown gel, and its molecular weight was 578 based on the ESI/MS ion peaks at *m/z* 579.1 [M + H]^+^ and 577.1 [M − H]^+^. The mass value implied the deduction of one hydroxyl group from the structure of compound **1**. It was also verified from the resonances of the B-ring aromatic protons that the structure has a para substitution at *δ*_H_ 7.90 (2H, *d*, *J* = 9.0, H-2′ and H-6′) and 6.92 (2H, *d*, *J* = 8.4, H-3′ and H-5′). All the other NMR data, besides those for the flavone B-ring, revealed the same structure as of compound **1**. Based on the spectroscopic data and literature, compound **3** was determined as apigenin-7-rutinoside [16]. 

Compound **4,** collected as a brown gel, had a molecular weight of 608, as determined by the ESI/MS ion peaks at *m/z* 609.2 [M + H]^+^ and 607.5 [M − H]^+^. Compared to the mass data for compound **3**, a methoxy group was added, and it was observed in the ^1^H and ^13^C NMR spectrum at *δ*_H_ 3.86 (3H, s, H-3′-OCH_3_) and *δ*_C_ 56.6 (C-3′-OCH_3_). Furthermore, from the anomeric carbon and proton peaks (*δ*_C_ 101.1 (C-1‴), 100.4 (C-1″), *δ*_H_ 5.03 (1H, *d*, *J* = 7.2, H-1″), 4.52 (1H, s, H-1‴)), and sugar moiety, resonances appeared from ppm: 3.13–3.62 in ^1^H and 66.6–76.8 in ^13^C NMR. Thus, the glycoside was determined as rutinoside, the same as that in compounds **1** and **3**. The aglycone was determined as chrysoeriol, attributable to one methoxy group attached to C-3′ in the B-ring, assigned based on the HMBC correlation as H-3′-OCH_3_ (*δ*_H_ 3.86)/C-3′ (*δ*_C_ 148.6). Therefore, compound **4** was identified as chrysoeriol-7-rutinoside [17].

Compounds **5** and **6** were isolated as brown powders, and their molecular weights were determined as 432 and 462, respectively, using ESI/MS. Both compounds had one glycoside, 7-*O*-glucoside, which was verified, similar to compound **2**. The aglycones of compounds **5** and **6** were apigenin [18] and chrysoeriol [19], similar to those of compounds **3** and **4**, respectively. Thus, compounds **5** and **6** were apigenin-7-*O*-glucoside [20] and chrysoeriol-7-*O*-glucoside, respectively.

Some of the isolated and identified flavone glycosides, **1**–**6**, have been reported in carrots. In a previous study in which several flavonoids, including flavones from DCL, were separated, as in this study, five compounds, except for compound **4,** were isolated, and compound **2** was shown to be the main flavone [21,22,23]. Furthermore, in other studies, compounds **1** and **2** were identified in the seeds and green leaves of carrots. Compounds **1** and **2** have been isolated from the methanol extracts of seeds at increased amounts, compared with the amounts isolated from DCL (30 mg of **1** and 15 mg of **2** from the seeds, and 10 mg of **1** and 7 mg of **2** from the DCL). However, the isolation and identification of compound **4** from DCL and the entire carrots have not yet been described. The structure of compound 4, chrysoeriol-7-rutinoside, from the DCL was elucidated for the first time. Compounds **2** and **5** were detected in DCL via LC/MS analysis in recent studies [24,25]. In other studies, all the compounds isolated in this study, **1–6**, were analyzed using ultra-high-performance liquid chromatography/ESI/MS [26,27]; however, there are no verified reports on the flavone content in DCL. Therefore, a standardized, reproducible analysis for quality control was necessary to enable the industrial utilization of DCL. We developed a reliable analysis employing the HPLC/UV system and using the flavones isolated in this study as standards.

### 2.2. Optimization of Analytical Conditions and Quantitation

To develop an accurate and reliable HPLC method for the flavones, **1**–**6**, and their hydrolyzed aglycones, **7**–**9**, identified in DCL, the sample extraction solvents, HPLC column resins, wavelengths, solvent gradient system, and column oven temperatures were optimized. MeOH, ethanol (EtOH), H_2_O, and their mixtures are commonly used to extract vegetable flavonoids and polar glycosides. Therefore, we examined the yield of compounds from DCL under several solvent conditions. The mixture of 70% EtOH in H_2_O was found to be the most appropriate solvent for extraction. The other solvents, such as 100% MeOH or EtOH, resulted in a poor yield of glycosides, and the flavones were poorly soluble in H_2_O. Furthermore, the aqueous MeOH solvent resulted in a lower yield than that obtained using the EtOH and H_2_O mixture, which was not attractive from the industrial perspective. Extraction in 70% EtOH, performed twice with ultra-sonication for 5 min, was found to be optimal. With regard to the optimal HPLC conditions, a YMC-Triart C18 column (5 µm, 4.6 mm × 250 mm) was used based on a typical ODS resin, which increased the resolution of the polar components. An important factor in quantitating flavonoids is UV absorbance; the maximum absorbance is generally observed at 300–350 nm. Therefore, the maximum absorbance of the nine standard compounds was investigated, and 346 nm was selected as the wavelength representative of all the standard compounds. A solvent gradient system comprising ACN (A) and H_2_O with 0.1% phosphoric acid as a buffer (B) was the most suitable for separating the standard compounds in the leaf extracts without impurities. The optimized gradient system was as follows: A/B = 13/83–22/78% (0–10 min), 22/78–23/77% (10–18 min), and 23/77–37/67% (18–35 min). The column oven was maintained at 40 °C, and the injection volume was 10 µL. Under these optimized conditions, the peaks for the nine standard compounds were observed at 13.7 (**1**), 14.5 (**2**), 16.4 (**3**), 17.2 (**4**), 17.9 (**5**), 20.1 (**6**), 27.3 (**7**), 30.5 (**8**), and 31.5 (**9**) min (Figure 2).

We employed the optimized conditions described above to compare flavone glycosides **1**–**6** and flavones **7**–**9** and assess the activities of DCL. Based on the analysis of five replicate samples of DCL and DCR, we found that the DCL contained all the standard compounds except **8**, and that the glycosides were the major components; the content of the flavone glycosides was as follows: 2.7378 ± 0.0295 mg/g of **1**, 6.6595 ± 0.0105 mg/g of **2**, 1.4797 ± 0.0150 mg/g of **3**, 2.7711 ± 0.0148 mg/g of **4**, 1.0465 ± 0.0110 mg/g of **5**, and 0.8607 ± 0.0128 mg/g of **6**. The content of flavones was as follows: 0.0593 ± 0.0004 of **7** and 0.2947 ± 0.0019 mg/g of **8**. The content of compounds **4** and **6** in the DCR was quantified as 0.0682 ± 0.0009 of **4** and 1.0062 ± 0.0082 mg/g of **6**; however, the other compounds were not detected (Table 1). DCR, along with its rich and diverse nutrients, contains carbohydrates as approximately 10.6% of the main energy source and 2.71–4.53% of its constituent free-sugars [28]. Thus, these glycosides were predictable; moreover, the results were different from our expectations. This fact is also evidenced by previous studies, in which the glycosides identified by us were reported from DCL, whereas only luteolin and apigenin were identified from DCR [29]. Only two chrysoeriols (**4** and **6**) were obtained from the BuOH extract layer of the DCR extract using the same extraction and fractionation process as for DCL. The structure of the compounds was verified using spectrometric data and by comparison with the standard compounds.

### 2.3. Validation of the Assay Method 

From the stock of the mixture of the 9 standard compounds (0.5 mg/mL), 13 sequentially diluted solutions (0.00001–0.2 mg/mL) were obtained and analyzed in 3 replicates. Each calibration curve implied good linearity with correlation coefficients (*r^2^*) greater than 0.9998 (Table 2).

The DCL samples were spiked with three concentrations of the standard mixture (1.2, 2.4, and 3.8 mg/mL). The accuracy and precision were evaluated by determining the recovery (%) and RSD values of the samples in inter- and intra-day experiments using four replicates (Table 3). The accuracy of recovery ranged from 95.15 ± 1.25 to 101.70 ± 0.19% for **1**, from 94.89 ± 0.45 to 103.10 ± 1.95% for **2**, from 100.67 ± 0.75 to 102.77 ± 0.81% for **3**, from 96.42 ± 0.92 to 99.17 ± 1.59% for **4**, from 100.57 ± 1.27 to 104.03 ± 1.10% for **5**, from 101.05 ± 1.42 to 104.74 ± 1.60% for **6**, from 100.75 ± 0.87 to 102.70 ± 0.23% for **7**, from 102.74 ± 0.18 to 105.97 ± 0.27% for **8**, and from 100.97 ± 0.84 to 105.77 ± 1.66% for **9**. The RSD values (%), indicative of precision, were estimated as 0.18–1.31% for **1**, 0.31–1.89% for **2**, 0.39–1.51% for **3**, 0.56–1.88% for **4**, 0.46–1.26% for **5**, 0.39–1.52% for **6**, 0.22–0.96% for **7**, 0.09–0.45% for **8**, and 0.48–1.57% for **9**. These recovery and RSD values represent the accuracy and precision of the nine DCL sample standards. 

The limit of detection (LOD) and limit of quantitation (LOQ) values were evaluated based on the average standard deviation (SD) value with eight replicates from the nine low-concentration standard compounds (0.05 µg/mL), which substituted the five low-concentration regression curves (0.01–0.1 µg/mL). The LOD and LOQ values were estimated by multiplying each SD by 3.3 and 10, respectively. Based on this, the LOD of each standard was estimated as 0.0376 for **1**, 0.0398 for **2**, 0.0292 for **3**, 0.302 for **4**, 0.0538 for **5**, 0.0375 for **6**, 0.0179 for **7**, 0.0558 for **8**, and 0.0351 µg/mL for **9**. The LOQ was 0.0999 for **1**, 0.1064 for **2**, 0.0744 for **3**, 0.0774 for **4**, 0.1491 for **5**, 0.0996 for **6**, 0.0401 for **7**, 0.1549 for **8**, and 0.0923 for **9**. 

### 2.4. Deglycosylation of Flavone Glycosides by Enzyme Treatment

In nature, most flavonoids exist as nonabsorbable and biologically inactive glycosides. The presence of a glycoside moiety in a flavonoid increases its molecular weight, rendering it unusable by the human body, and reduces its activity [30]. The bioavailability of glucosides is increased by hydrolysis of the glucose moiety using enzymatic conversion [31]. The effect of the enzymatic complexes, Viscozyme L containing cellulase (Viscozyme) and Pectinase (Pectinex), on flavone glycosides from DCL has been examined.

There are reports confirming that the antioxidant activities of flavone glycosides from DCL increase, along with the total flavonoid and phenol content, when treated with enzymes [32]. Therefore, in this study, the deglycosylation of flavone glycosides from DCL by treatment with Viscozyme L and Pectinex was assessed, and the hydrolyzed compounds were analyzed using the HPLC method. 

After treatment of the DCL sample with the enzymes at a concentration of 0.1% to verify the efficiency of the enzyme reaction over time, the decomposition and generation of the compounds were monitored for up to 24 h. All the flavone glycosides (**1**–**6**) in DCL were gradually hydrolyzed upon treatment with Viscozyme L (Table 4). Their content was determined to be 0.03 ± 0.00 for **1**, 0.22 ± 0.00 for **2**, 0.02 ± 0.00 for **3**, 0.59 ± 0.10 for **4**, 0.06 ± 0.03 for **5**, and 0.07 ± 0.04 mg/mL for **6**. Using Pectinex, the final content was 0.21 ± 0.01 for **1**, 2.02 ± 0.12 for **2**, 0.10 ± 0.01 for **3**, 0.94 ± 0.05 for **4**, 0.40 ± 0.02 for **5**, and 0.26 ± 0.01 mg/mL for **6**. In contrast, the deglycosylated flavones were gradually produced until 24 h had passed, and their final content was 6.34 ± 0.09 for **7**, 0.40 ± 0.05 for **8**, and 1.72 ± 0.03 mg/g for **9** using Viscozyme L and 5.43 ± 0.09 for **7**, 0.35 ± 0.01 for **8**, and 1.45 ± 0.09 mg/g for **9** using Pectinex. Although there were no significant differences in these enzymes, as illustrated in the conversion rate graph over response time (Figure 3), considering the data for the converted content comprehensively, Viscozyme L was more efficient than Pectinex at the same concentration.

Based on the law of mass conservation, to verify the loss of compounds in the samples during enzyme treatment, the molecular weight of each flavone glycoside was estimated as a percentage of the molecular weight when the sugar was degraded. Furthermore, by substituting this in the flavone glycoside before and after the enzyme reaction, the content of relatively pure flavone was estimated, and the total content of each flavone was compared. Thus, the three groups of the flavone glycoside aglycones and content of the converted flavones, including luteolin, apigenin, and chrysoeriol, were compared. The total content of the luteolin group, including compounds **1**, **2,** and **7**, was 6.41 mg/mL at 0.5 h and 5.50 mg/mL at 24 h after treatment with Viscozyme L. The total content of the apigenin group, including compounds **3**, **5**, and **8**, was 1.36 mg/mL at 0.5 h and 0.45 mg/mL at 24 h after treatment with Viscozyme L. In the chrysoeriol group, including compounds **4**, **6**, and **7**, the total content was 2.31 mg/mL at 0.5 h and 2.05 mg/mL at 24 h after the Viscozyme L treatment. Upon converting the reaction contents to % values, the luteolin, apigenin, and chrysoeriol groups showed values of 85.8, 33.1, and 88.7%, respectively. These values validated the conversion yield of the flavone glycosides in the sample, allowing a more realistic correlation with the results of sample activation following the enzyme reaction (Figure 2 and Figure 3). 

The treatment of 1 g DCL extract in 70% ethanol with Viscozyme L and Pectinex resulted in 15.91 mg/g and 14.83 mg/g of total flavonoids, respectively. There was no significant difference in the conversion efficiency between these enzymes, but Viscozyme L showed a higher conversion rate to flavonoids than Pectinex. A similar trend was observed for the enzymatic conversion of flavone glycosides isolated from DCL.

### 2.5. Anti-Inflammatory Activities of the Samples 

We determined the effects of DCR, DCL, and enzyme-treated DCL on the expression of TNF-*α* and IL-2 in stimulated human T lymphocyte cells. TNF-*α* and IL-2 are crucial to several immune-mediated inflammatory diseases [33,34,35,36]. As illustrated in Figure 4, cells treated with phorbol 12-myristate 13-acetate (PMA)/phytohemagglutinin-L (PHA) expressed higher levels of TNF-*α* and IL-2 mRNAs than non-treated cells. The treatment with DCR and DCL extracts considerably decreased the expression of TNF-*α* and IL-2 under PMA and PHA stimulation. In addition, DCL extract exhibited a stronger inhibitory activity than DCR. When treated with Viscozyme and Pectinex, the DCL had a higher mRNA inhibitory effect than that of the DCR and untreated DCL. Furthermore, Jurkat cells stimulated with PMA/PHA were used to verify the effects of DCR and DCL on TNF-α and IL-2 expression. Similar to the mRNA expression results, the DCL displayed more potent inhibitory effects than the DCR. The inhibitory effect of enzyme-treated DCL on the expression of TNF-*α* and IL-2 was higher than that of DCR. These results indicated that TNF-α was differentially regulated at the transcriptional level, and the release of cytokines was associated with the enzyme treatment. The enzyme treatment of DCL resulted in a stronger inhibitory effect than that in DCR or DCL without enzyme treatment.

## 3. Materials and Methods

### 3.1. General Procedures

The solvents used in extraction and separation were EtOH (>95% pure), extra pure-grade *n*-hexane (HEX), ethyl acetate (EA), and *n*-butanol (BuOH). They were purchased from Dae-Jung Chemicals and Metals (Gyeonggi-do, Korea). In addition, we used triple-distilled water (H_2_O). All analytical-grade solvents, ACN, and H_2_O were obtained from J. T. Baker (Phillipsburg, NJ, USA). The analytical-grade buffer and phosphoric acid were purchased from Sigma-Aldrich (St. Louis, MO, USA). The enzymes Viscozyme L (cellulolytic enzyme mixture) and Pectinex (pectinase from *Aspergillus aculeatus*) were purchased from Sigma-Aldrich. The open-column chromatography was performed using silica gel (Kieselgel 60, 70–230 mesh, Merck & Co., Rahway, NJ, USA). The preparative HPLC was conducted using an LC-Forte/R model (YMC, Tokyo, Japan). The NMR data (^1^H-NMR, ^13^C-NMR, ^1^H-^1^H COSY, HMBC, and HMQC) were obtained from the 600 MHz NMR, JNM-ECA600 (Jeol, Tokyo, Japan) with DMSO-*d*_6_ and CD_3_OH as solvents and TMS as the internal standard. The ESI-MS data were analyzed using Bruker maXis 4G (Bruker, Madison, WI, USA) mass spectrometers in positive- and negative-ion modes. The HPLC system was equipped with a Shimadzu LC-20A Prominence Series system and a quaternary pump (LC-20AD), vacuum degasser (DGU-20A3R), autosampler (SIL-20A), column oven (CTO-20A), and photodiode-array detector (SPD-M20A). Chromatographic data were processed using the LabSolutions Multi PDA software (Shimadzu Corporation, Kyoto, Japan).

### 3.2. Sample Preparation 

Carrot seeds were purchased from a local market in Daejeon, Korea, in March 2018 and were cultivated in the Korea Institute of Oriental Medicine (KIOM) for 3 months after being sown in July. Samples used in the separation and analysis were harvested in October 2018. To secure the samples, an expert committee from the KIOM was consulted. The validated samples of the DCR and DCL were separated and deposited as a standard plant in KIOM (KIOM DCL 181010 and KIOM DCR 181011). Each sample was cleaned, and the DCR were chopped into small pieces. Thereafter, the DCR and DCL were dried at 45 °C for 3 days in a drying oven. 

### 3.3. Separation and Isolation Procedures

The dried DCL (1 kg) were extracted thrice with 5 L of 70% EtOH at 80 °C for 24 h. The extracted solutions were evaporated under vacuum, and 207.25 g of the 70% EtOH extract was obtained. Furthermore, 185 g of the total extract was suspended in H_2_O (1.5 L) and sequentially partitioned into HEX (3 L × 4), EtOAc, (3 L × 4), and BuOH (3 L × 4). All four separated layers were concentrated using vacuum evaporation. The BuOH extract (18.1 g) was selected based on HPLC chromatogram patterns and results of the biological activity for separation and isolation of the compounds. Column chromatography was conducted using a semi-preparative HPLC with a YMC Triat C18 column (5 µm, 150 mm × 21.20 mm i.d.) at 254 and 320 nm, a flow rate of 15 mL/min, and a gradient system with H_2_O:acetonitrile (ACN) (90:10–75:35 *v*/*v*) to separate and identify the active components. After separating the BuOH extract, six fractions (fr. B1–B6) were obtained. The highest content was obtained from fraction B6 (7.2 g), and six main peaks were observed in the chromatogram. Thus, for the next separation step, fraction B6 (1.0 g) was fractionated using semi-preparative HPLC equipped with a YMC Triat C18 column (5 µm, 150 mm × 21.20 mm i.d.). The HPLC conditions included UV detection at 254 and 320 nm and isocratic flow of the solvent (H_2_O:ACN = 85:15) at 15 mL/min. Six compounds, namely **1** (274 mg), **2** (66.8 mg), **3** (14.8 mg), **4** (27.8 mg), **5** (10.6 mg), and **6** (8.7 mg), were isolated. The purity of all these compounds was verified to be >95% using HPLC analysis. 


**Luteolin-7-rutinoside (Scolymoside) (1)**


ESI-MS ion peaks at *m*/*z* 595.0 [M + H]^+^ and 592.9 [M − H]^+^. ^1^H NMR data in CD_3_OD (600 MHz): δ_H_ 7.37(2H, s, H-2′ and H-6′), 6.90 (1H, s, H-5′), 6.70 (1H, s, H-9), 6.56 (1H, s, H-3), 6.48 (1H, s, H-7), 5.01 (1H, s, H-1″), 4.69 (1H, s, H-1‴), 4.03–3.31 (10H, Overlap, sugar moiety), and 1.16 (3H, d, *J* = 6.0, 5‴-CH_3_). ^13^C NMR data in CD_3_OD (600 MHz): δ_C_ 182.6 (C-4), 165.6 (C-2), 163.3 (C-8), 161.5 (C-6), 157.5 (C-10), 149.8 (C-4′), 145.6 (C-3′), 122.2 (C-1′), 119.3 (C-6′), 115.6 (C-5′), 113.0 (C-2′), 105.8 (C-5), 102.9 (C-3), 100.8 (C-1‴), 100.3(C-1″), 99.8 (C-7), 94.8 (C-9), 76.5 (C-5″), 75.8(C-3″), 73.4 (C-3‴), 72.7 (C-4‴), 71.1 (C-2‴), 70.8 (C-2″), 70.0 (C-5‴), 68.5 (C-4‴), 66.2 (C-6″), and 16.5 (C-5‴-CH_3_).


**Luteolin-7-*O*-glucoside (Cynaroside) (2)**


ESI-MS ion peaks at *m*/*z* 449.38 [M + H]^+^ and 447.80 [M − H]^+^. ^1^H NMR data in DMSO-*d*_6_ (600 MHz): δ_H_ 7.40 (2H, d, *J* = 8.4, H-2′ and H-6′), 6.88 (1H, br s, H-5′), 6.75 (1H, s, H-9), 6.69 (1H, s, H-3), 6.40 (1H, d, *J* = 1.8, H-7), 5.04 (1H, d, *J* = 7.2, H-1″), and 3.69–3.15 (6H, Overlap, sugar moiety). ^13^C NMR data in DMSO-*d*_6_ (600 MHz): δ_C_ 182.4 (C-4), 165.0 (C-2), 163.4 (C-8), 161.6 (C-6), 157.4 (C-10), 150.5 (C-4′), 146.3 (C-3′), 121.9 (C-1′), 119.7 (C-6′), 116.5 (C-5′), 114.1 (C-2′), 105.9 (C-5), 103.7 (C-3), 100.5 (C-1″), 100.1 (C-7), 95.3 (C-9), 77.7 (C-5″), 76.9 (C-3″), 73.7 (C-2″), 70.1 (C-4″), and 61.2 (C-6″).


**Apigenin-7-rutinoside (3)**


ESI-MS ion peaks at *m*/*z* 579.1 [M + H]^+^ and 577.1 [M − H]^+^. ^1^H NMR data in DMSO-*d*_6_ (600 MHz): δ_H_ 7.90 (2H, d, *J* = 9.0, H-2′ and H-6′), 6.92 (2H, d, *J* = 8.4, H-3′ and H-5′), 6.80 (1H, s, H-3), 6.73 (1H, d, *J* = 1.8, H-9), 6.41 (1H, d, *J* = 1.8, H-2), 5.02 (1H, d, *J* = 7.8, H-1″), 4.52 (1H, s, H-1‴), 3.82 (1H, d, *J* = 10.2, H-6″), 3.62 (1H, s, H-2‴), 3.56 (1H, m, H-3‴), 3.34 (7H, Overlap, sugar moiety), and 1.04 (3H, d, *J* = 6.6, 5‴-CH_3_). ^13^C NMR data in DMSO-*d*_6_ (600 MHz): δ_C_ 182.5 (C-4), 167.9 (C-2), 163.4 (C-8), 161.8 (C-4′), 161.7 (C-6), 157.4 (C-10), 129.1 (C-2′ and C-6′), 121.6 (C-1′), 116.6 (C-3′ and C-5′), 105.9 (C-5), 103.6 (C-3), 101.0 (C-1‴), 100.5 (C-1″), 100.1 (C-7), 95.3 (C-9), 76.8 (C-5″), 76.2 (C-3″), 73.6 (C-3‴), 72.6 (C-4‴), 71.3 (C-2‴), 70.9 (C-2″), 68.9 (C-5‴), 66.6 (C-4″), and 18.3 (5‴-CH_3_).


**Chrysoeriol-7-rutinoside (4)**


ESI-MS ion peaks at *m*/*z* 609.2 [M + H]^+^ and 607.5 [M − H]^+^. ^1^H NMR data in DMSO-*d*_6_ (600 MHz): δ_H_ 7.54 (2H, m, H-2′ and H-6′), 6.93 (1H, d, *J* = 8.4, H-5′), 6.92 (1H, s, H-3), 6.76 (1H, d, *J* = 2.4, H-9), 6.42 (1H, d, *J* = 2.4, H-7), 5.03 (1H, d, *J* = 7.2, H-1″), 4.52 (1H, s, H-1‴), 3.86 (3H, s, H-3′-OCH_3_), 3.81 (1H, d, *J* = 10.2, H-6″α), 3.62–3.13 (9H, Overlap, sugar moiety), and 1.04 (3H, d, J = 6.6, 5‴-CH_3_). ^13^C NMR data in DMSO-*d*_6_ (600 MHz): δ_C_ 182.5 (C-4), 164.8 (C-2), 163.4 (C-8), 161.7 (C-6), 157.4 (C-10), 151.4 (C-4′), 148.6 (C-3′), 121.9 (C-1′), 121.1 (C-6′), 116.4 (C-5′), 110.8 (C-2′), 105.9 (C-5), 104.0 (C-3), 101.1 (C-1‴), 100.4 (C-1″), 100.0 (C-7), 95.5 (C-9), 76.8 (C-5″), 76.2 (C-3″), 73.6 (C-3‴), 72.6 (C-4‴), 71.3 (C-2‴), 70.1 (C-2″), 68.9 (C-5‴), 66.6 (C-4″), 56.6 (C-3′-OCH_3_), and 18.3 (5‴-CH_3_).


**Apigenin-7-*O*-glucoside (5)**


ESI-MS ion peaks at *m*/*z* 433.0 [M + H]^+^ and 430.9 [M − H]^+^. ^1^H NMR data in DMSO-*d*_6_ (600 MHz): δ_H_ 7.91 (2H, d, *J* = 8.4, H-2′ and H-6′), 6.90 (2H, d, *J* = 2.4, H-3′ and H-5′), 6.81 (1H, s, H-3), 6.79 (1H, d, *J* = 1.2, H-9), 6.41 (1H, d, *J* = 2.4, H-7), 5.03 (1H, d, *J* = 7.8, H-1″), 3.68 (1H, d, *J* = 11.4, H-2″), and 3.33–3.15 (5H, Overlap, sugar moiety). ^13^C NMR data in DMSO-*d*_6_ (600 MHz): δ_C_ 182.5 (C-4), 164.8 (C-2), 163.5 (C-8), 161.9(C-4′), 161.6 (C-6), 157.5 (C-10), 129.1 (C-2′ and C-6′), 121.5 (C-1′), 116.5 (C-3′ and C-5′), 105.9 (C-5), 103.6 (C-3), 100.1 (C-1″), 95.4 (C-7), 77.7 (C-5″), 77.0 (C-3″), 73.7 (C-2″), 70.1 (C-4″), and 61.2 (C-6″).


**Chrysoeriol-7-*O*-glucoside (6)**


ESI-MS ion peaks at *m*/*z* 463.0 [M + H]^+^ and 461.1 [M − H]^+^. ^1^H NMR data in DMSO-*d*_6_ (600 MHz): δ_H_ 7.54(2H, d, *J* = 9.6, C-2′ and C-6′), 6.91 (2H, d, *J* = 5.4, C-3′ and C-5′), 6.82 (1H, s, H-9), 6.41 (1H, d, *J* = 1.8, H-6), 5.02 (1H, d, *J* = 7.2, H-1″), 3.85 (3H, s, 3′-OCH_3_), 3.70 (1H, d, *J* = 10.8, H-6″α), and 3.46–3.16 (4H, Overlap, sugar moiety). ^13^C NMR data in DMSO-*d*_6_ (600 MHz): δ_C_ 181.2 (C-4), 163.4 (C-2), 162.1 (C-8), 160.3 (C-6), 156.1 (C-10), 150.1 (C-4′), 147.2 (C-3′), 120.6 (C-1′), 119.7 (C-6′), 115.0 (C-5′), 109.5 (C-2′), 104.6 (C-5), 102.6 (C-3), 99.2 (C-9), 98.7 (C-1″), 94.2 (C-7), 76.5 (C-5″), 75.7 (C-3″), 72.4 (C-2″), 68.9 (C-4″), 59.9 (C-6″), and 55.2 (3′-OCH_3_).

### 3.4. Preparation of Standards and Samples for HPLC Analysis

Among the nine standards used for HPLC analysis, six glycoside compounds, namely luteolin-7-rutinoside (**1**), luteolin-7-*O*-glucoside (**2**), apigenin-7-rutinoside (**3**), chrysoeriol-7-rutinoside (**4**), apigenin-7-*O*-glucoside (**5**), and chrysoeriol-7-*O*-glucoside (**6**), with >95% purity were isolated and identified in this study. The remaining three aglycones, apigenin (**7**), luteolin (**8**), and chrysoeriol (**9**), with >98% purity, were obtained from Sigma-Aldrich. The mixture of the nine compounds was diluted to 0.5 mg/mL in 70% EtOH and stored at 4 °C as a stock solution. Furthermore, it was consecutively diluted to 18 concentrations (0.00001–0.2 mg/mL) as working solutions before HPLC analysis. 

Dried DCL and DCR were powdered, and 1 g of each was extracted twice with 10 mL of 70% EtOH for 5 min with sonication to prepare the HPLC samples. The extracted sample solutions were filtered using a disposable syringe filter (0.22 µm, 25 mm, CA syringe fitter) from Futecs Co., LTD (Daejeon, Korea). For preparing enzyme-treated DCL samples, 1 g of dried powder was extracted with 3 mL of H_2_O containing 0.1% (*v*/*w*) of Viscozyme L or Pectinex, in a water bath set at 50 °C, in five successive treatments: 30 min, 1 h, 3 h, 8 h, and 24 h. EtOH (7 mL extra-pure reagent) was added and the mixture was vortexed for 1 min to deactivate the enzymatic reaction at each time point. All the deactivated samples were sonicated for 5 min and passed through the same disposable syringe filter (0.22 µm, 25 mm, CA syringe fitter) before injecting into the HPLC system.

### 3.5. Validation of the Developed Analytical Method

The HPLC analysis was validated in terms of linearity, accuracy, precision, LOD, and LOQ following the guidelines of the International Conference on Harmonization (ICH). The linearity was established by evaluating *r*^2^ (correlation coefficient) values for the calibration curves generated using 10 serial concentrations. The precision of the analysis was examined by intermediate evaluation using measurements of the intra- and inter-day variability. The intra-day variability was determined by analyzing the sample solution during one of the study days (24 h). In contrast, the inter-day variability was assessed over four days by injecting the sample solutions five times daily. Relative standard deviation (RSD) values were estimated for the retention time and peak area in five experiments. RSD was a measure of precision. Recovery tests were performed to evaluate accuracy in the sample solution spiked with each standard compound. Recovery rates were determined by estimating the mean recovery (%) of the standards from the spiked extract solutions vs. the non-spiked extract sample. LOD and LOQ were determined using the signal-to-noise ratio (S/N), in which S/N ratios of **3** for LOD and 10 for LOQ were used.

### 3.6. Cell Culture and Reagents

Human T lymphocyte cells, Jurkat cells, were obtained from American Type Culture Collection (ATCC). The cells were maintained in Roswell Park Memorial Institute (RPMI) 1640 medium containing 10% fetal bovine serum, 100 U/mL penicillin, and 100 mg/mL streptomycin and maintained in a humidified atmosphere containing 5% CO_2_ at 37 °C. PMA and PHA were purchased from Sigma-Aldrich.

### 3.7. Reverse Transcription and Real-Time Polymerase Chain Reaction

Total RNA was extracted using the RNeasy Mini Kit (Qiagen, Hilden, Germany), following the manufacturer’s instructions. cDNA (1 µg) was synthesized from total RNA using the Thermo Scientific RevertAid First Strand cDNA Synthesis Kit (Thermo Fisher Scientific, Waltham, MA, USA) and amplified by real-time reverse transcription polymerase chain reaction (RT-PCR) using AmpliTaq Gold DNA polymerase and quantitative real-time PCR. The cDNA was amplified using Premix ExTaq (TaKaRa, Shiga, Japan) with SYBR Premix EX Taq (TaKaRa) via the BI PRISM 7500HT Sequence Detection System (Applied Biosystems, Waltham, MA, USA). The primers were synthesized by Macrogen Inc. (Seoul, Korea). Actin expression was used as the control. The following custom-designed primers were used: 5′-GTCTCCTCTGACTTCAACAGCG-3′ and 3′-ACCACCCTGTTGCTGTAGCCAA-5′ for *GAPDH*; 5′-CTCTTCTGCCTGCTGCACTTTG-3′ and 3′-ATGGGCTACAGGCTTGTCACTC-5′ for TNF-α: 5′-AGAACTCAAACCTCTGGAGGAAG-3′ and 3′-GCTGTCTCATCAGCATATTCACAC-5′ for IL-2.

### 3.8. Quantitation of TNF-α and IL-2 in Cell Culture Supernatants Using ELISA

TNF-α and IL-2 levels in the cell supernatants were measured using a Human TNF-α and IL-2 ELISA kit (Abcam, Cambridge, England), according to the manufacturer’s instructions. The OD value was measured at 540 nm using a multi-mode microplate reader (Berthold Technology, Calmbacher, Germany).

### 3.9. Statistical Analysis

Data are presented as means ± SD. Paired Student’s *t*-test was used to compare the groups and ANOVA with Tukey’s test was used for multiple comparison using the PRISM software (v6.0; GraphPad Software, San Diego, CA, USA). *p*-values < 0.05 were considered significant.

## 4. Conclusions

In this study, six flavone glycosides were isolated from 70% EtOH DCL extract, and their structures were identified via NMR and MS spectroscopy. Among the isolated compounds **1–6**, the structure of compound **4**, chrysoeriol-7-rutinoside, from the DCL has been elucidated for the first time. Structural elucidations, DCL quality control, and the reproducibility and accuracy of the HPLC/UV analytical method for the six isolated flavone glycosides along with their three aglycones were fully validated for the first time. From the validation study, all standard compounds in the DCL samples were estimated with accuracy and precision, as confirmed by the recoveries of 94.89–105.97% and RSDs < 1.89. The deglycosylation of flavone glycosides from DCL by treatment with Viscozyme L and Pectinex was assessed, and the hydrolyzed compounds were analyzed using the HPLC/UV method. The deglycosylated flavones were gradually produced until 24 h, and their final content was 6.34 ± 0.09 (apigenin), 0.40 ± 0.05 (luteolin), and 1.72 ± 0.03 mg/g (chrysoeriol) after treatment with Viscozyme L and 5.43 ± 0.09 (apigenin), 0.35 ± 0.01 (luteolin), and 1.45 ± 0.09 mg/g (chrysoeriol) after treatment with Pectinex. Upon converting the reaction contents to %, the luteolin, apigenin, and chrysoeriol groups showed values of 85.8, 33.1, and 88.7%, respectively.

The effects of the extracts of DCR, DCL, and enzyme-treated DCL extract on the expression of TNF-α and IL-2 in human T lymphocyte cells were investigated using PMA/PHA stimulation. These extracts considerably inhibited the expression of TNF-α and IL-2. The inhibition of TNF-α (*p* < 0.001) and IL-2 (*p* < 0.0001) expression by enzyme-treated DCL extract was higher than the inhibition by DCR extract and untreated DCL extract. 

The new discovery pertaining to the compounds present in DCL may provide valuable data for standardization of raw materials, which is essential for the development of functional foods and drugs and may also provide valuable information for its general use as food.

## Figures and Tables

**Figure 1 molecules-28-04291-f001:**
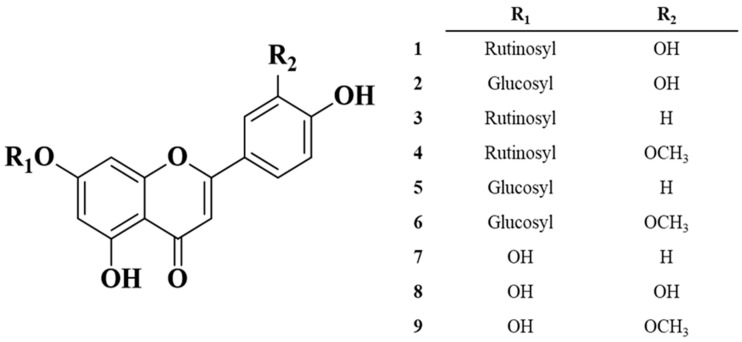
Chemical structures of isolates from carrot leaves (DCL; 70% extracts and hydrolyzed compounds as standards).

**Figure 2 molecules-28-04291-f002:**
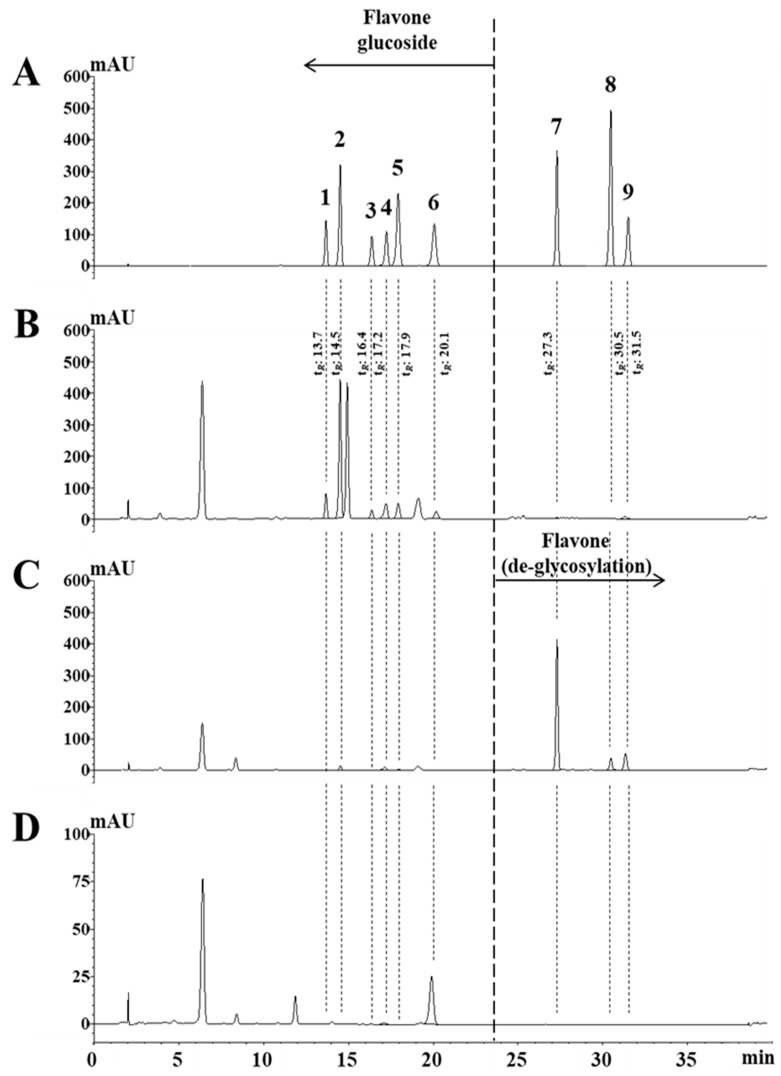
HPLC chromatograms of the seven standard compounds ((**A**), 0.1 mg/mL), the DCL ((**B**), 20 mg/mL), the DCL after deglycosylation of flavone glycosides by enzyme treatment ((**C**), 20 mg/mL), and the DCR ((**D**), 20 mg/mL).

**Figure 3 molecules-28-04291-f003:**
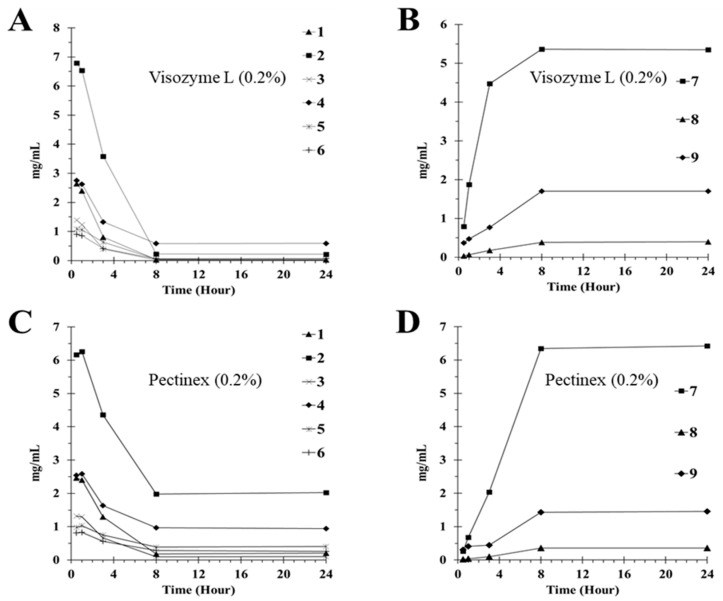
Effect of reaction time on the enzymatic deglycosylation of standard compounds from the DCL. (**A**): compound **1**–**6** using Viscozyme L (0.2%), (**B**): compound **7**–**10** using Viscozyme L (0.2%), (**C**): compound **1**–**6** using Pectinex (0.2%), and (**D**): compound **7**–**9** using Pectinex (0.2%).

**Figure 4 molecules-28-04291-f004:**
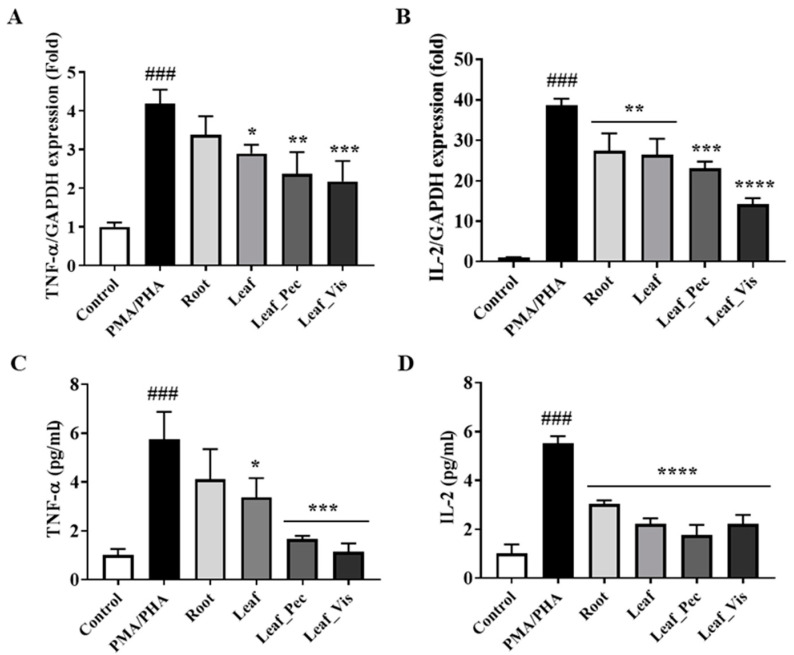
Effect of DCR, DCL, and enzyme-treated DCL on the transcript levels of TNF-α and cytokines under various stimulatory conditions. (**A**,**B**) The mRNA levels of TNF-α and IL-2 were assayed using RT-PCR. Jurkat cells were stimulated using PMA (100 ng/mL)/PHA (10 ng/mL) and treated with various extracts, as indicated for 3 h. The mRNA levels of TNF-α and IL-2 were determined using qRT-PCR and normalized to those of GAPDH. (**C**,**D**) TNF-α and IL-2 levels were measured using an enzyme-linked immunosorbent assay (ELISA). Cells were stimulated with PMA (100 ng/mL)/PHA (10 ng/mL) and cotreated with various extracts, as indicated for 24 h. After simulation, the supernatants were collected for ELISA. The values are presented as the mean ± SD for three independent experiments. ### *p* < 0.0001 vs. control and **** *p* < 0.0001, *** *p* < 0.001, ** *p* < 0.005, * *p* < 0.05 vs. PMA/PHA.

**Table 1 molecules-28-04291-t001:** Content of the target compounds in the DCL and DCR samples (mg/g).

Compound	Content (*n* = 5)	
	DCL	DCR
**1**	2.7378 ± 0.0295 *^b^*	N.D.
**2**	6.6595 ± 0.0105	N.D.
**3**	1.4797 ± 0.0150	N.D.
**4**	2.7711 ± 0.0148	0.0682 ± 0.0009
**5**	1.0465 ± 0.0110	N.D.
**6**	0.8607 ± 0.0128	1.0062 ± 0.0082
**7**	0.0593 ± 0.0004	N.D.
**8**	N.D. *^a^*	N.D.
**9**	0.2947 ± 0.0019	N.D.

*^a^* Not detected, *^b^* Standard error (mg/g).

**Table 2 molecules-28-04291-t002:** Linearity of the standard compounds.

Compound	t*_R_* (min)	Linear Range (mg/mL)	Equation (Linear Model) *^a^*	*r*^2^ *^b^*
**1**	13.7	0.0002–0.2	y = 12,536,160.4409x − 3665.4389	0.9998
**2**	14.5	0.0002–0.2	y = 29,391,463.6658x − 10,361.4359	0.9998
**3**	16.4	0.0002–0.2	y = 10,086,940.3468x − 2969.9097	0.9998
**4**	17.2	0.0002–0.2	y = 12,664,623.4452x − 3672.0329	0.9998
**5**	17.9	0.0002–0.2	y = 30,244,089.0439x − 8460.5572	0.9998
**6**	20.1	0.0002–0.2	y = 20,910,201.0414x − 6370.9364	0.9998
**7**	27.3	0.0002–0.2	y = 31,733,472.2675x − 10,337.5869	0.9998
**8**	30.5	0.0002–0.2	y = 52,480,014.7485x − 16,388.1711	0.9998
**9**	–	0.0002–0.2	y = 18,491,893.6013x − 5697.0590	0.9998

*^a^* y: peak area at 324 nm; x: standard concentration (mg/mL). *^b^ r^2^*: coefficient of determination with 13 indicated points in the calibration curves.

**Table 3 molecules-28-04291-t003:** Intra- and inter-day accuracy and precision of the standard compounds in DCL samples with four replicates.

Compound	Spiked (mg/mL)	Inter-Day		Intra-Day	
		Recovery (%, *n* = 4)	RSD *^a^* (%)	Recovery (%, *n* = 4)	RSD (%)
**1**	1.2	95.15 ± 1.25	1.31	97.10 ± 0.28	0.29
	2.4	100.73 ± 0.89	0.89	101.70 ± 0.19	0.18
	3.8	97.44 ± 0.33	0.33	97.26 ± 0.26	0.27
**2**	1.2	103.10 ± 1.95	1.89	96.38 ± 0.30	0.31
	2.4	102.71 ± 1.19	1.16	102.51 ± 1.22	1.19
	3.8	94.89 ± 0.45	0.48	95.44 ± 0.72	0.76
**3**	1.2	101.55 ± 1.54	1.51	102.77 ± 0.81	0.79
	2.4	100.67 ± 0.75	0.75	101.41 ± 0.82	0.81
	3.8	101.16 ± 0.43	0.42	100.97 ± 0.39	0.39
**4**	1.2	97.51 ± 1.72	1.76	99.17 ± 1.59	1.60
	2.4	96.94 ± 1.82	1.88	98.98 ± 1.42	1.44
	3.8	97.24 ± 0.55	0.56	96.42 ± 0.92	0.96
**5**	1.2	103.72 ± 0.72	0.70	104.03 ± 1.10	1.06
	2.4	100.57 ± 1.27	1.26	101.23 ± 0.47	0.46
	3.8	102.35 ± 0.48	0.47	102.62 ± 0.58	0.56
**6**	1.2	104.74 ± 1.60	1.52	102.52 ± 0.40	0.39
	2.4	101.60 ± 1.53	1.51	102.41 ± 1.43	1.40
	3.8	102.36 ± 1.13	1.11	101.05 ± 1.42	1.14
**7**	1.2	102.33 ± 0.98	0.96	101.51 ± 0.45	0.45
	2.4	100.75 ± 0.87	0.86	100.95 ± 0.22	0.22
	3.8	102.70 ± 0.23	0.23	102.48 ± 0.41	0.40
**8**	1.2	105.59 ± 0.47	0.45	105.97 ± 0.27	0.26
	2.4	102.74 ± 0.18	0.18	102.89 ± 0.17	0.17
	3.8	103.86 ± 0.09	0.09	103.86 ± 0.16	0.15
**9**	1.2	105.77 ± 1.66	1.57	102.43 ± 1.16	1.13
	2.4	101.15 ± 0.48	0.48	100.97 ± 0.84	0.83
	3.8	103.64 ± 0.61	0.58	104.51 ± 0.51	0.49

*^a^* RSD: relative standard deviation.

**Table 4 molecules-28-04291-t004:** Quantification of individual compounds in the DCL after deglycosylation with different enzyme treatment times.

Enzyme	Compound	Compound Contents (mg/g)				
		0.5 h	1 h	3 h	8 h	24 h
**Vis** *^a^*	**1**	2.65 ± 0.01 *^c^*	2.40 ± 0.08	0.80 ± 0.07	0.03 ± 0.00	0.03 ± 0.00
	**2**	6.80 ± 0.02	6.53 ± 0.23	3.58 ± 0.32	0.23 ± 0.00	0.22 ± 0.00
	**3**	1.39 ± 0.01	1.23 ± 0.04	0.39 ± 0.03	0.02 ± 0.00	0.02 ± 0.00
	**4**	2.75 ± 0.01	2.63 ± 0.09	1.33 ± 0.12	0.59 ± 0.01	0.59 ± 0.10
	**5**	1.09 ± 0.01	1.05 ± 0.03	0.63 ± 0.06	0.05 ± 0.00	0.06 ± 0.03
	**6**	0.90 ± 0.01	0.86 ± 0.02	0.41 ± 0.04	0.06 ± 0.00	0.07 ± 0.04
	**7**	0.79 ± 0.01	1.88 ± 0.06	4.47 ± 0.40	5.96 ± 0.12	6.34 ± 0.09
	**8**	0.03 ± 0.00	0.07 ± 0.00	0.18 ± 0.01	0.39 ± 0.01	0.40 ± 0.05
	**9**	0.37 ± 0.00	0.48 ± 0.01	0.77 ± 0.07	1.71 ± 0.04	1.72 ± 0.03
	**1**	2.47 ± 0.14	2.40 ± 0.11	1.30 ± 0.02	0.19 ± 0.01	0.21 ± 0.01
**Pec** *^b^*	**2**	6.16 ± 0.35	6.06 ± 0.26	4.53 ±0.04	1.98 ± 0.19	2.02 ±0.12
	**3**	1.32 ± 0.07	1.29 ± 0.05	0.67 ± 0.01	0.09 ± 0.01	0.10 ± 0.01
	**4**	2.54 ± 0.14	2.39 ± 0.10	1.63 ± 0.01	0.96 ± 0.09	0.94 ±0.05
	**5**	0.98 ± 0.05	0.82 ± 0.04	0.75 ± 0.01	0.39 ± 0.04	0.40 ± 0.02
	**6**	0.81 ± 0.05	0.83 ± 0.03	0.56 ± 0.00	0.29 ± 0.02	0.26 ± 0.01
	**7**	0.26 ±0.02	0.68 ± 0.03	2.04 ± 0.03	5.34 ± 0.63	5.43 ±0.09
	**8**	0.02 ± 0.00	0.03 ± 0.00	0.10 ± 0.00	0.35 ± 0.03	0.35 ± 0.01
	**9**	0.31 ± 0.02	0.41 ± 0.02	0.44 ± 0.01	1.43 ± 0.15	1.45 ± 0.09

*^a^* Viscozyme L, *^b^* Pectinex, ^*c*^ Standard error (mg/g).

## Data Availability

All relevant data are presented in the manuscript. Raw data will be provided upon reasonable request.

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
