# Peer review of "Efficiency of the Enzymatic Conversion of Flavone Glycosides Isolated from Carrot Leaves and Anti-Inflammatory Effects of Enzyme-Treated Carrot Leaves"

_molecules, 2023, doi:10.3390/molecules28114291_

Round 1

Reviewer 1 Report

I must congratulate the authors on the clear and concise way in which the article is written, with extremely important supporting analytical techniques such as mass spectrometry, and with all results being well presented and discussed, revealing a high level of scientific maturity.

In my opinion, the article should be accepted in its current form.

Author Response

Thank you for your review on the thesis.

I will study hard so that I can write a good thesis in the future.

Reviewer 2 Report

The manuscript "Efficiency of the enzymatic conversion of flavone glycosides isolated from carrot leaves and anti-inflammatory effects of enzyme-treated carrot leaves" is a well-constructed study that addresses an underutilized source of natural products, carrot leaves. The manuscript does a good job of describing the background, methods, results, and conclusions. The manuscript describes a very sound methods and presents the data and conclusions in a very easy to read fashion. The authors are very detailed in their descriptions of a wide range of experiments from the isolation and extraction, structure elucidation, quantification, enzymatic processing, to cell-based assays. I believe readers, particularly those with interests in natural products and/or flavones will find this manuscript informative and interesting. I suggest publication with minor edits to language as follows:

1.     In line 10, it defines the acronym for D. carota leaves as DGL. Why is it DGL and not DCL based on the genus and species of the carrot?

2.     In line 12, “developing plants” do you mean “processing vegetables”?

3.     In line 28, I would move “owing to their rich nutrient content” to beginning of sentence.

4.     I would remove new paragraph in line 55 and then delete the sentence “We suggest the possibility of industrial application of carrot leaves”

5.     In line 60, I would define the enzyme type for Viscozyme and Pectinex.

6.     On page 3, I would replace the term “resonances” with peaks or signals in all instances.

7.     In line 172, there is a random 3 after the period.

8.     In line 182, “Table 1” is in bold in the text. This is the only instance of it being in bold in text in the paper. Please make uniform.

9.     In lines 285-288, The figure legend should be all together and not double spaced out, so information is properly formatted.

10.  In the paragraph from lines 305-320, the authors are repeatedly missing the term “extracts” after “DGL”, “DGR”, or “enzyme-treated DGL”. I believe in all instances the authors are referring to extracts, but only use term extract in lines 310 and 311.

The english is overall very good. there are some minor fixes to grammar and/or word choice. Please see comments on other document.

Author Response

  1. In line 10, it defines the acronym for D. carota leaves as DGL. Why is it DGL and not DCL based on the genus and species of the carrot?

- Answer: I thank for your comment. I changed DGL to DCL.

  1. In line 12, “developing plants” do you mean “processing vegetables”?

- Answer: In line 12, 'developing plants' means 'processing vegetables', and was changed as follows.

generally treated as waste while processing vegetables for wide industrial availability.

  1. In line 28, I would move “owing to their rich nutrient content” to beginning of sentence.

- Answer: In line 28was changed as follows. Owing to their rich nutrient contents, carrots have long been used as antifungal, antibacterial, and nephroprotective agents [3, 4], including essential oils.

  1. I would remove new paragraph in line 55 and then delete the sentence “We suggest the possibility of industrial application of carrot leaves”

- Answer: I thank for your comment. I removed new paragraph in line 55 and deleted the sentence "We propose industrial application potential of carrot leaves" as you suggested.

  1. In line 60, I would define the enzyme type for Viscozyme and Pectinex.

- Answer: In line 60 was changed as follows. In addition, the flavone glycosides were treated with two commercial food-grade enzymes of Viscozyme and Pectinex, and three hydrolyzed flavones were isolated and quantitatively analyzed (Figure 1).

  1. On page 3, I would replace the term “resonances” with peaks or signals in all instances.

- Answer: I thank for your comment. "Resonance" is a technical term for NMR analysis. As a substitute for "Resonance", I think signal rather than peak is more appropriate.

  1. In line 172, there is a random 3 after the period.

- Answer: I thank for your comment. I removed a random 3 after the period.

  1. In line 182, “Table 1” is in bold in the text. This is the only instance of it being in bold in text in the paper. Please make uniform.

- Answer: I thank for your comment. "Table 1" has been changed to normal font.

  1. In lines 285-288, The figure legend should be all together and not double spaced out, so information is properly formatted.

- Answer: I thank for your comment. In lines 285-288 were changed as follows.

Figure 3. Effect of reaction time on the enzymatic deglycosylation of standard compounds from the DCL.  A: compound 1–6 using Viscozyme L (0.2%), B: compound 7–10 using Viscozyme L (0.2%), C: compound 1–6 using Pectinex (0.2%), and D: compound 7–9 using Pectinex (0.2%).

  1. In the paragraph from lines 305-320, the authors are repeatedly missing the term “extracts” after “DGL”, “DGR”, or “enzyme-treated DGL”. I believe in all instances the authors are referring to extracts, but only use term extract in lines 310 and 311.

- Answer: I thank for your comment. In lines 305-320 were changed as follows.

We determined the effects of the extracts of DCR, DCL, and enzyme-treated DCL extract on the expression of TNF-α and IL-2 in stimulated human T lymphocyte cells. TNF-α and IL-2 are crucial to several immune-mediated inflammatory diseases [33-37]. As illustrated in Figure 4, cells treated with phorbol 12-myristate 13-acetate (PMA)/phytohemagglutinin-L (PHA) ex-pressed higher levels of TNF-α and IL-2 mRNAs than did non-treated cells. Treatment with DCR and DCL extracts considerably decreased the expression of TNF-α and IL-2 under PMA and PHA stimulation. In addition, DCL extract exhibited stronger inhibitory activity than DCR extract. When treated with Viscozyme and Pectinex, the DCL extract had a higher mRNA inhibitory effect than that of the DCR extract and untreated DCL extract. Furthermore, Jurkat cells stimulated with PMA/PHA were used to verify the effects of the extracts of DCR and DCL on TNF-α and IL-2 expression. Similar to the mRNA expression results, the DCL extract displayed more potent inhibitory effects than the DCR extract . The inhibitory effect of enzyme-treated DCL extract on the ex-pression of TNF-α and IL-2 was higher than that of DCR extract. These results indicated that TNF-α was differentially regulated at the transcriptional level, and the release of cytokines was associated with the enzyme treatment. The enzyme treatment of DCL extract resulted in a stronger inhibitory effect than that in the extracts of DCR or DCL without enzyme treatment.

Reviewer 3 Report

thus work is presented in a very structured way. In my opinion, the results of this work could be a starting point for applications in the industrial field of food products. I only ask the authors to indicate the results considered to be statistically significant from the results at the level of the tables.

Author Response

(The authors gave the same response as above.)
